# Immunoinformatic Approaches to Identify Immune Epitopes and Design an Epitope-Based Subunit Vaccine against Emerging Tilapia Lake Virus (TiLV)

Sk Injamamul Islam [1,*], Sarower Mahfuj [1], Md. Ashraful Alam [2], Yeasmin Ara [3], Saloa Sanjida [4] and Moslema Jahan Mou [5]

1    Department of Fisheries and Marine Bioscience, Faculty of Biological Science, Jashore University of Science and Technology, Jashore 7408, Bangladesh; sa.mahfuz@gmail.com
2    Rural Development Academy (RDA), Bogura 5842, Bangladesh; ashhstu019@gmail.com
3    Department of Fisheries Management, Faculty of Fisheries, Hajee Mohammad Danesh and Technology University, Dinajpur 5200, Bangladesh; yeasminara3@gmail.com
4    Department of Environmental Science and Technology, Faculty of Applied Science and Technology, Jashore University of Science and Technology, Jashore 7408, Bangladesh; saloa.sanjida94@gmail.com
5    Department of Genetic Engineering and Biotechnology, Faculty of Biological Science, Jashore University of Science and Technology, Jashore 7408, Bangladesh; moslemajahanmou@gmail.com
*    Correspondence: 6378506331@student.chula.ac.th; Tel.: +880-1728387603

**Abstract:** Tilapia tilapinevirus, known worldwide as tilapia lake virus (TiLV), is a single-stranded RNA virus that belongs to the Amnoonviridae family. The virus attacks the fish species' external and internal organs, such as the eyes, brain, and liver. Syncytial cells develop in the liver cells of infected fish, which are characterized by widespread hepatocellular necrosis and karyolytic nuclei. It is a highly infectious virus that spreads both horizontally and vertically. Despite these devastating complications, there is still no cure or vaccine for the virus. Therefore, a vaccine based on epitopes developed using immunoinformatics methods was developed against TiLV in fish. The putative polymerase basic 1 (PB1) gene was used to identify immunodominant T- and B-cell epitopes. Three probable epitopes were used to design the vaccine: CTL, HTL, and LBL. Testing of the final vaccine revealed that it was antigenic, non-allergenic, and has improved solubility. Molecular dynamics simulation revealed significant structural compactness and binding stability. Furthermore, the computer-generated immunological simulation indicated that immunization might stimulate real-life immune responses following injection. Overall, the findings of the study imply that the designed epitope vaccine might be a good option for prophylaxis for TiLV.

**Keywords:** TiLV; T-cell epitopes; immunoinformatics; molecular docking; molecular dynamics simulation

## 1. Introduction

Tilapia lake virus (TiLV) is a growing virus that kills cultured (*Oreochromis* spp. and hybrids) and hybrid tilapia in Asia, Africa, Central America, and South America. Globally, tilapia fry, juveniles, and adults are reported to have the TiLV virus, resulting in 100% mortality and significant economic losses since 2014. Despite not yet being designated as a viral family, this orthomyxo-like virus includes 10 negative-sense RNA segments wrapped inside a membrane-bound nucleocapsid. The viral particle has a diameter of 55 to 100 nanometers and is typically circular. TiLV is vulnerable to organic solvents such as ether and chloroform because it has an outer lipid membrane. Even though the length of time TiLV may live outside of its host is unknown, tests using tilapia demonstrate that it can spread horizontally and persist in both freshwater and brackish water. Previous research found the putative polymerase basic 1 (PB1) gene in TiLV infected tilapia in Thailand [1]. Identities in nucleotides and amino acids were quite high between Thailand and Israel for

TiLVs (95.18–99.10%). For the putative PB1 gene of Thailand's TiLV, both nucleotide and amino acid identity were high (99.61–100%) [1].

Cytotoxic T-lymphocyte epitopes of fish were discovered using in vivo experiments and a peptide database of the overlapping pathogens [2]; T-cell epitopes of CD4+ were detected in a diverse range of species [3] via epitopes mapping [4]. To fight against pathogens from outside and to boost immunity or immune response, a vaccine can be used as a weapon by infiltrating pathogens, reducing toxicity, or eliminating pathogens from the body. Vaccines may help prevent future epidemics of natural microorganisms associated with viruses, according to previous research [5]. The rapid development of secure, accurate, easy, cost-effective, trustworthy, and in silico development of epitope-based vaccines against viruses allows for the fast establishment of innate immunity against the directed antigen. In the postgenomic era, epitope-based vaccinations were effective against human viruses in stimulating an immune response [6–10]. Due to the lack of proper information regarding major histocompatibility complex I and II with HLA (Human leukocyte antigen), the in silico approach in fish was not invented before now [11,12]; however, the current study of fish species produced data that will allow immunoinformatics approaches to be used [13–15]. A recent study found both the class I and II of MHC in cod and tilapia in an in vivo experiment. As a consequence, the peptide, which has strong binding interactions with the previously reported HLA found in tilapia, might be used as an effective vaccine against specific diseases in tilapia [11,16]. Recently, an in silico method was used against *Streptococcus agalactiae, Edwardsiella tarda*, and *Flavobacterium columnare*, the three most important fish pathogens, to predict the best antigenic and immunogenic peptides [17–19]. Researchers suggest that computer-assisted solutions will become increasingly effective in managing fish infections in the coming years [20,21]. Therefore, the primary goal of this study is to find multi-epitopes from the most effective antigenic protein to combat TiLV.

## 2. Methods

Figure 1 illustrates the flow chart of the methods of the study.

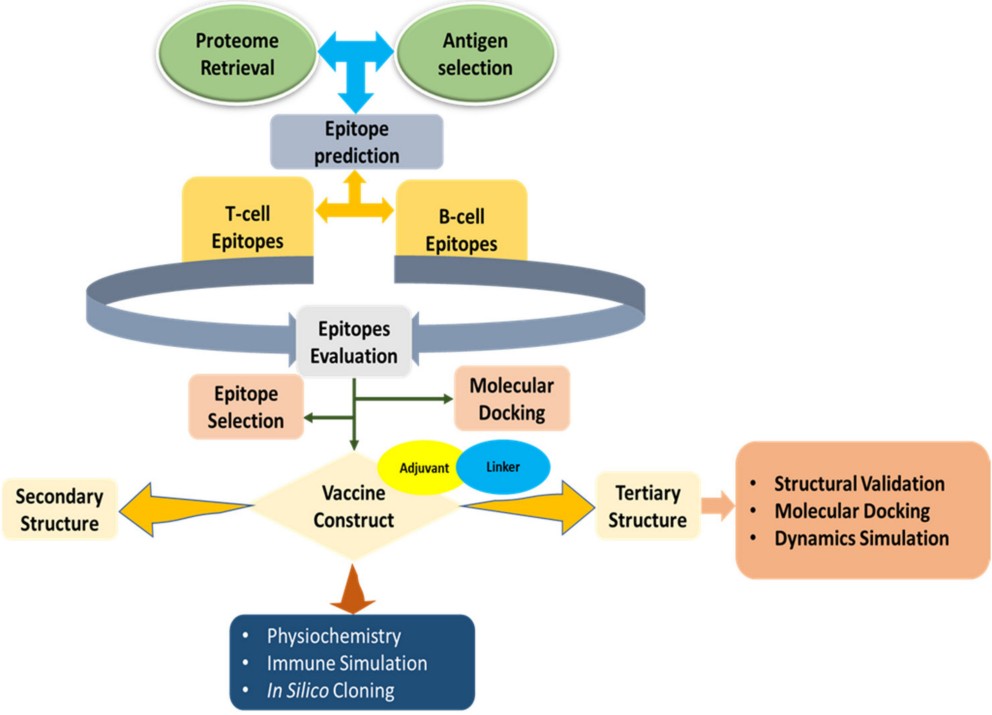

**Figure 1.** Depicts an architectural flow chart.

### 2.1. Antigen Selection and Retrieval of the Proteome

We used the NCBI database to find accessible tilapia lake virus (TiLV) proteomes for antigen selection. Putative polymerase basic 1 (PB1) is a critical protein and the perfect target antigen for the design of a TiLV vaccine because it permits viruses to enter the cell wall of the host [1]. Owing to its direct participation in pathogenesis, the TiLV PB1 was studied for multi-epitope vaccine creation. Following the isolation of the PB1, the virus's preferred protein sequences were retrieved as FASTA files. The protective antigens were assessed using a threshold value of 0.4 on the VaxiJen v2.0 [22] and the ANTIGENpro server [23].

### 2.2. CTLs Epitope Prediction and Evaluation

CTLs (cytotoxic T-lymphocytes) are one type of immune cell capable of infecting other cells with viruses or bacteria [24]. Viruses enter viral cells very quickly and contribute to the defense response of the host. To anticipate CTL epitopes, the PB1 protein sequence was input into the NetCTL v1.2 server [25]. VaxiJen v2.0 [22], MHC class I immunogenicity [26], ToxinPred [27], and AllerTop v2.0 [28] servers were utilized to determine the epitopes' immunogenicity. All of the projections were made within the servers' default parameters.

### 2.3. HTLs Epitope Prediction and Assessment

HTLs (helper T-lymphocytes) cause the immune system to destroy infectious pathogens by identifying foreign antigens and activating B- and cytotoxic T-cells [29]. The IEDB's binding allele prediction tool for MHC class II epitopes was used to calculate the HTL epitopes. HTL epitopes were selected utilizing the consensus method, with a percentile of 5% [30]. The antigenicity and cytokine-inducing capabilities of the predicted epitopes, IFNc, IL4, and IL10, were further investigated. The antigenicity score was calculated using VaxiJen v2.0; IFNc, IL4, and IL10 characteristics were anticipated by applying default parameters using IFNepitope [31], IL4pred [31], and IL10pred [32] servers, respectively.

### 2.4. Epitopes of Linear B-Lymphocytes: Prediction and Evaluation

For enhancing antibody-mediated and humoral responses, B-cell epitopes are essential. They contain amino acids that attach to antibody molecules and stimulate the immune system's antibody production [33]. Therefore, the linear B-lymphocyte (LBL) epitopes were predicted in this study using the iBCE-EL server with default parameters [34]. Antigenicity testing was performed using the VaxiJen v2.0 server, toxicity was predicted using the ToxinPred v2.0 server, and lastly, the allergenicity of the epitopes was predicted using the AllerTop v2.0 server.

### 2.5. Molecular Docking and Peptide Modeling

For the docking analysis with the receptor alleles, 3D modeling of the selected CTLs and HTLs epitopes was performed using the PEP-FOLD v3.0 server. The operation was carried out using the sOPEP sorting system with 200 simulations [35]. CTL epitopes were chosen based on HLA-B*3501 (PDB:1A9E) and HLA-A*0201 (PDB:3MGO), whereas HTL epitopes were chosen based on HLA binding allele analysis of the epitope-wise HLA binding allele. The Protein Data Bank (PDB) was utilized to retrieve the 3D crystal structures of alleles [36], and BIOVIA Discovery Studio 2020 was used to process all the structures. It is necessary to create a grid box around the neighboring HLA alleles' active sites before the molecular docking analysis using the AutoDock tool and AutoDock Vina script [37,38]. The co-crystal ligands were set as a positive control to properly analyze and evaluate the binding efficacy of the epitopes [39]. Finally, to visualize the binding interactions of epitopes with neighboring alleles, the PBDsum server and BIOVIA Discovery Studio 2017 were used.

### 2.6. Development of an Epitope-Based Vaccination

To design and enhance the efficacy of the vaccine, all the epitopes were selected from three different classes (CTL, HTL, and LBL), and connected with specific adjuvant and

linkers. TLR4 agonist was employed as an adjuvant in this study to achieve excellent translation and synthesis of the designed vaccine [40,41]. To enhance the immunogenicity of the vaccine candidate, 50S ribosomal protein L7/L12 (NCBI ID: P9WHE3) was tested as an adjuvant. In addition to the EAAAK bifunctional linker, a peptide length-dependent cleavage mechanism was described that can break apart two weakly interacting b domains over a range of peptide lengths. AAY linkers were used to connect the chosen CTL epitopes, GPGPG linkers were used to connect selected HTL epitopes, and KK linkers were used to connect the LBL epitopes [33]. As a proteasome cleavage site, the AAY linker can be exploited to alter protein stability and immunogenicity, as well as to enhance epitope presentation [42]. To facilitate immune processing, a "junctional epitope" was avoided, while the bi-lysine KK linker helps to retain the vaccine construct's discrete immunogenic characteristics.

### 2.7. Evaluation of Physicochemical and Immunological Factors

The essential features of a protein are described by its physiochemistry. Physicochemical properties of the vaccine were predicted using ProtParam to understand the vaccine's essence [43]. For the evaluation of immunological parameters, we used the VaxiJen v2.0 server [22], MHC-I immunogenicity server [26], AllerTop server [28], and SOLpro server [23].

### 2.8. 3D Structure Prediction

The SOPMA server [44] and PSIPRED v4.0 server [45] discovered the construct's secondary characteristics. SOPMA's prediction accuracy is greater than 80% [44]. To better understand the new vaccine's composition quality, 2D structural parameters were extracted and analyzed.

### 2.9. Validation, 3D Structure Refinement, and Homology Modeling

The designed vaccine sequence was uploaded to the RaptorX server [46]. This server can generate the most exact 3D protein structure model using a cutting-edge algorithm [46]. This online tool can predict and calculate the C-score, TM-score value, RMSD, and top five models of a certain protein sequence. The PDB file that was picked based on the C-score was used to save the created 3D structure. On the server, the C-score runs from –5 to 2, with a higher value suggesting a more reliable protein model. The GalaxyRefine online web-based platform was used to refine the 3D structure of the vaccine. The CASP10 refining method was used to run the server [47]. RMSD, energy scores, and overall quality scores were obtained from this server. The modified structure was downloaded on the basis of the lowest energy scores and the maximum RMSD values. Finally, PyMOL v2.3.4 was used to visualize the refined 3D structure of the vaccine [48]. After that, the model quality was evaluated by applying the Ramachandran plot score and Z-score value, which reflect standard deviations from the mean. The Ramachandran plots were evaluated using PROCHECK, an application that analyzes the most allowed and banned areas of amino acid sequences; and the Z-score plots were studied using ProSA-web [49].

### 2.10. Disulfide Engineering of the Toccine

In order to move forward and begin docking analysis, the designed model must be stable. Proteins with disulfide bonds have a geometrically stable structure. For the intended vaccination, Disulfide by Design 2.0 [50] was utilized to generate such connections.

### 2.11. Studies on Molecular Docking

Molecular docking helps analyze the binding residues between the protein-receptor (ligand) complexes. The designed vaccine model was used as a ligand and the receptor was the TLR4 (PDB ID: 4G8A), downloaded from the PDB database for Molecular Docking using the ClusPro v2.0 server [51]. The initial stage in preparing the receptor was to separate the associated ligand from the receptor, which was accomplished by eliminating water

and other substances. The PyMOL v2.3.4 software [48] was used for all of these processes. Ultimately, to analyze the binding residues and interacting surfaces from the vaccine-TLR4 complex, the PDB sum online server and BIOVIA Discovery Studio 2017 were used.

### 2.12. Molecular Dynamics Simulation

The interaction stability of the candidate of a selected compound to the specific protein was examined via 50 ns molecular dynamic simulations (MDS) [52]. This MDS was run in Schrödinger 2020-3 under the Linux framework using the "Desmond v6.3 Program" to investigate the receptor-ligand complex's thermodynamic stability [53]. An orthorhombic periodic boundary box shape with a box distance of 10 Å was allocated to both sides of the reservoir to retain a specific volume within the reservoir. The system was reduced and relaxed using the default protocol presented inside the Desmond module with OPLS 2005 force field settings after generating the solvated system comprising protein in complex with the ligand [53]. In the protein preparation wizard, the protein is preprocessed utilizing Epik (pH: 7.0 ± 2.0) and PROPKA to add hydrogens, create disulfide linkages, and fulfill the side chains. The model system used simulation period = 50 ns, trajectory intervals = 50 ps, overall number of frames = 1000, ensembles class = NPT, warmth = 300 K, and one atmospheric (1.01325 bar) pressure for simulation. Furthermore, the simulation was run for 50 ns, and the trajectories were examined for RMSF, RMSD, and protein secondary structural components to determine the vaccination complex's stability.

### 2.13. Simulation of Immune Response

The designed vaccine sequence was submitted to the C-IMMSIM v10.1 server to evaluate the specific immune response of the vaccine [54]. Two dosages, with a minimum gap of 30 days, were applied in this study [55]. Three injections, with time steps of 1, 84, and 168, were administered using an in silico approach, with one time step equaling 8 h in real life. All other simulation parameters were kept at their usual levels, with the simulation step value set at 300 max.

## 3. Results
### 3.1. Antigenicity Prediction

The TiLV proteomes that were obtained contained potential proteins (PB1). Based on the antigenicity server, Vaxijen predicted that the antigenicity score for PB1 was 0.4824 and the ANTIGENpro server predicted that the antigenicity score was 0.607. The selected putative protein was 519 amino acids long and has the entry number QJD15206.1 in GenBank.

### 3.2. Prediction of Potential CTL Epitopes

This study predicted 47 CTL epitopes from PB1 protein with a length of nine amino acids. Out of 47 CTL epitopes, 19 epitopes were identified with antigenic, immunogenic, non-toxic, and non-allergenic characteristics. The top three leading epitopes were chosen, based on the highest antigenicity score for this study, to design the final vaccine sequence (Table 1).

**Table 1.** CTL epitopes chosen for the final vaccination.

| Epitope | C-Score | Antigenicity | Immunogenicity | Toxicity | Allergenicity |
|---------|---------|--------------|----------------|----------|---------------|
| YTATASAEQ | 0.8770 | 0.5345 | Positive | Negative | Negative |
| GTTDRFLSF | 0.5838 | 0.7173 | Positive | Negative | Negative |
| VSAVYTATA | 0.5343 | 0.6773 | Positive | Negative | Negative |

### 3.3. Prediction of Best HTL Epitopes

A total of 299 HTL epitopes with a length of 15 amino acids long were predicted using the IEDB server to design the vaccine. Among them, the top 12 epitopes were found to

have IL4, IL10, and IFNc inducing features. Finally, the top 3 epitopes among 12 epitopes were selected for the final construction based on their antigenic scores (Table 2).

**Table 2.** HTL epitopes were selected for the final vaccination.

| Epitope | Antigenicity | IFN$_\gamma$ | IL4 | IL10 | Toxicity | Allergenicity |
|---|---|---|---|---|---|---|
| SLKKSYISVASLEIN | 1.0072 | Positive | Inducer | Inducer | Negative | Negative |
| LKKSYISVASLEINS | 0.8496 | Positive | Inducer | Inducer | Negative | Negative |
| LSLKKSYISVASLEI | 1.2590 | Positive | Inducer | Inducer | Negative | Negative |

IFN$_\gamma$: Interferon-gamma; IL4: Interleukin 4; IL10: Interleukin 10.

### 3.4. Potential LBL Epitopes

A preliminary examination discovered ten LBL epitopes, each measuring 12 amino acids long. Three epitopes were discovered to be antigenic, non-toxic, and non-allergenic after more study (Table 3).

**Table 3.** LBL epitopes chosen for the final vaccination.

| Epitope | Probability | Antigenicity | Allergenicity | Toxicity |
|---|---|---|---|---|
| LRDQERGKPKSR | 0.8157 | 1.9529 | Negative | Negative |
| RDQERGKPKSRA | 0.7908 | 1.7915 | Negative | Negative |
| DQERGKPKSRAI | 0.7332 | 1.3894 | Negative | Negative |

### 3.5. Epitopes with Neighboring Alleles Molecular Docking Study

The docking method was used to ensure that selected epitopes were effective in binding their HLA alleles. Table 4 lists the epitopes, together with their docking alleles, binding affinities, interaction, and hydrogen-bonding residues. The binding affinities of CTL epitopes were between −7.1 and −9.0 kcal/mol, while HTL epitopes were between −6.9 and −7.0 kcal/mol. Figure 2 demonstrates the top interacting CTL (TLTSEFLDF) and HTL (PTVSLPYNEVRIHFK) epitopes along with the tabulated details. The best CTL epitope generated 18 hydrogen bonds in total, 14 of which were classical interactions with the active site residues Asp9, Glu63, Lys66, Arg69, Asn77, Asn77, Lys80, Tyr84, Tyr99, Thr143, Lys146, Trp147, and Glu152. The best HTL epitope, on the other hand, connected with Asp9, Ser24, Glu63, Lys66, Arg69, Arg69, Tyr99, Glu152, Glu152, and Gln155 residues, forming 12 hydrogen bonds, including 10 classical interactions.

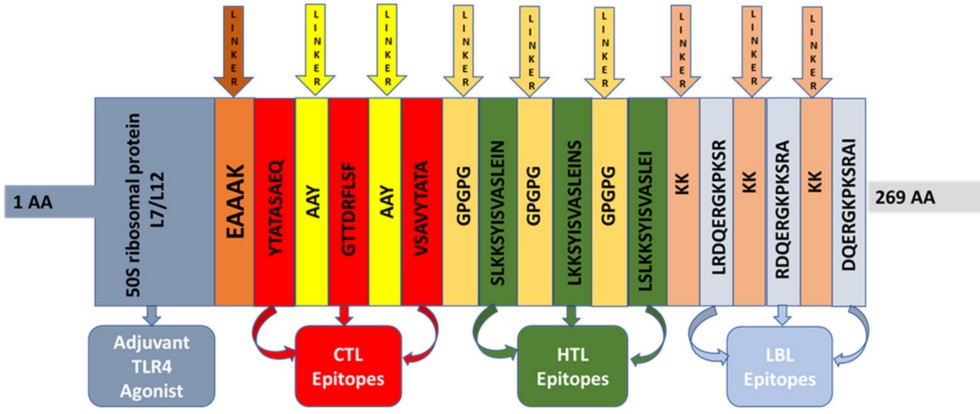

**Figure 2.** The vaccine design for the formed multi-epitope vaccine. AYY linkers were employed to connect CTL epitopes, GPGPG linkers were used to connect HTL epitopes, and KK linkers were used to connect LBL epitopes. The adjuvant and the first CTL epitope were linked using EAAAK linkers.

**Table 4.** Interactions and binding affinities between specific epitopes and HLA alleles.

| T-Cell Epitope | HLA Allele | Epitope Affinity (kcal/mol) | Control Affinity (kcal/mol) | Number of Hydrogen Bonds (CHB) | Residues Involved in CHB Networks (n) |
|---|---|---|---|---|---|
| YTATASAEQ | HLA-A*0201 | −7.2 | −9.2 | 9(7) | Gln69, Trp149, Thr7, Ile8, Met19, Ile1, Ala2, Ile7, Tyr74 (9) |
| GTTDRFLSF | HLA-A*0201 | −7.1 | −8.2 | 9(8) | Lys80, Tyr84, Lys146, Val2, Thr7, Val9, Lys66, Asn77, Thr143 (9) |
| VSAVYTATA | HLA-B*3501 | −9.0 | −8.2 | 18(14) | Tyr7, Arg2, Asp9, Glu63, Lys66, Arg69, Asn77, Asn77, Lys80,Tyr84, Tyr99, Thr143, Lys146, Trp147, Glu15, Glu152, Tyr159, Tyr171 (18) |
| SLKKSYISVASLEIN | DRB1*03:01 | −6.9 | −7.5 | 9(7) | Arg71, Thr77, Asn82, Ala12, Thr13, Val14, Val1, Glu6, Ser4 (9) |
| LKKSYISVASLEINS | DRB5*01:01 | −7.0 | −7.7 | 12(10) | Tyr7, Asp9, Asp9, Ser24, Glu63, Lys66, Arg69, Arg69, Tyr99, Glu152, Glu152, Gln155 (12) |
| LSLKKSYISVASLEI | DRB4*01:01 | −6.9 | −7.3 | 10(8) | Ser63, Glu85, Asn72, His328, Trp7, Ala15, Phe17, Tyr8, Ile17, Ile3 (10) |

### 3.6. Core Properties and Structure of a Vaccine

The vaccine was designed with the help of nine epitopes from three different classes (three CTL, three HTL, and three LBL) that had already been chosen. To enhance immunogenicity, adjuvant and linkers were employed before the design (Figure 2). To bind the adjuvant, EAAAK linker was used on the first CTL epitope to boost the immunogenicity of the vaccine. The final immunization was 268 amino acids in length (Figure 3).

MAKLSTDELLDAFKEMTLLELSDFVKKFEETFEVTAAAPVAVAAAGAAPAGAAVEAAEEQSEFDVILEAAGDKKIGVIKVVREIVSGLGLK
EAKDLVDGAPKPLLEKVAKEAADEAKAKLEAAGATVTVKEAAKYTATASAEQAAYGTTDRFLSFAAYVSAVYTATAGPGPGSLKKSYISVA
SLEINGPGPGLKKSYISVASLEINSGPGPGLSLKKSYISVASLEIKKLRDQERGKPKSRKKRDQERGKPKSRAKKDQERGKPKSRAI

**Figure 3.** Constructed vaccine sequence.

### 3.7. Prediction of the Physicochemical and Immunological Features of the Vaccine

The molecular weight of the construct was determined to be 28,256.39 Da. Other characteristics included the theoretical isoelectric point (pI) of 9.23, chemical formula $C_{1255}H_{2068}N_{340}O_{392}S_2$, instability index of 17.18, aliphatic index of 85.06, and grand average of hydropathicity of −0.292. The physicochemical characteristics and immunological effects of the construct were also evaluated. The antigenicity of the construct, for example, was 0.7053, but its immunogenicity was positive. The vaccine was similarly non-allergenic and soluble, receiving a score of 0.901123 on a scale of 1 (Table 5). To predict the secondary structural properties, α-helix, β-strand, and random coils were investigated using two different servers. In the design, the SOPMA server projected 48.70% α-helix, 14.13% β-strand, and 30.11% random coils (Table 6). The PSIPRED server similarly predicted 39.033% α-helix, 17.47% β-strand, and 43.494% random coils (Table 6 and Figure 4).

**Table 5.** Characteristics of the construct in terms of antigenicity, allergenicity, and physicochemical properties.

| Characteristics | Finding | Remark |
|---|---|---|
| Number of amino acids | 269 | Suitable |
| Aliphatic index of vaccine | 85.06 | Thermostable |
| Grand average of hydropathicity (GRAVY) | −0.292 | Hydrophilic |
| Antigenicity | 0.7053 | Antigenic |
| Immunogenicity | Positive | Immunogenic |
| Allergenicity | No | Non-allergen |
| Solubility | 0.901123 | Soluble |

**Table 6.** The vaccine's secondary structural characteristics.

| Characteristics | SOMPA Server | | PSIPRED Server | |
|---|---|---|---|---|
| | Amino Acid | % | AA | % |
| α-helix | 131 | 48.70% | 105 | 39.033% |
| β-strand | 38 | 14.13% | 47 | 17.47% |
| Random coil | 81 | 30.11% | 117 | 43.494% |

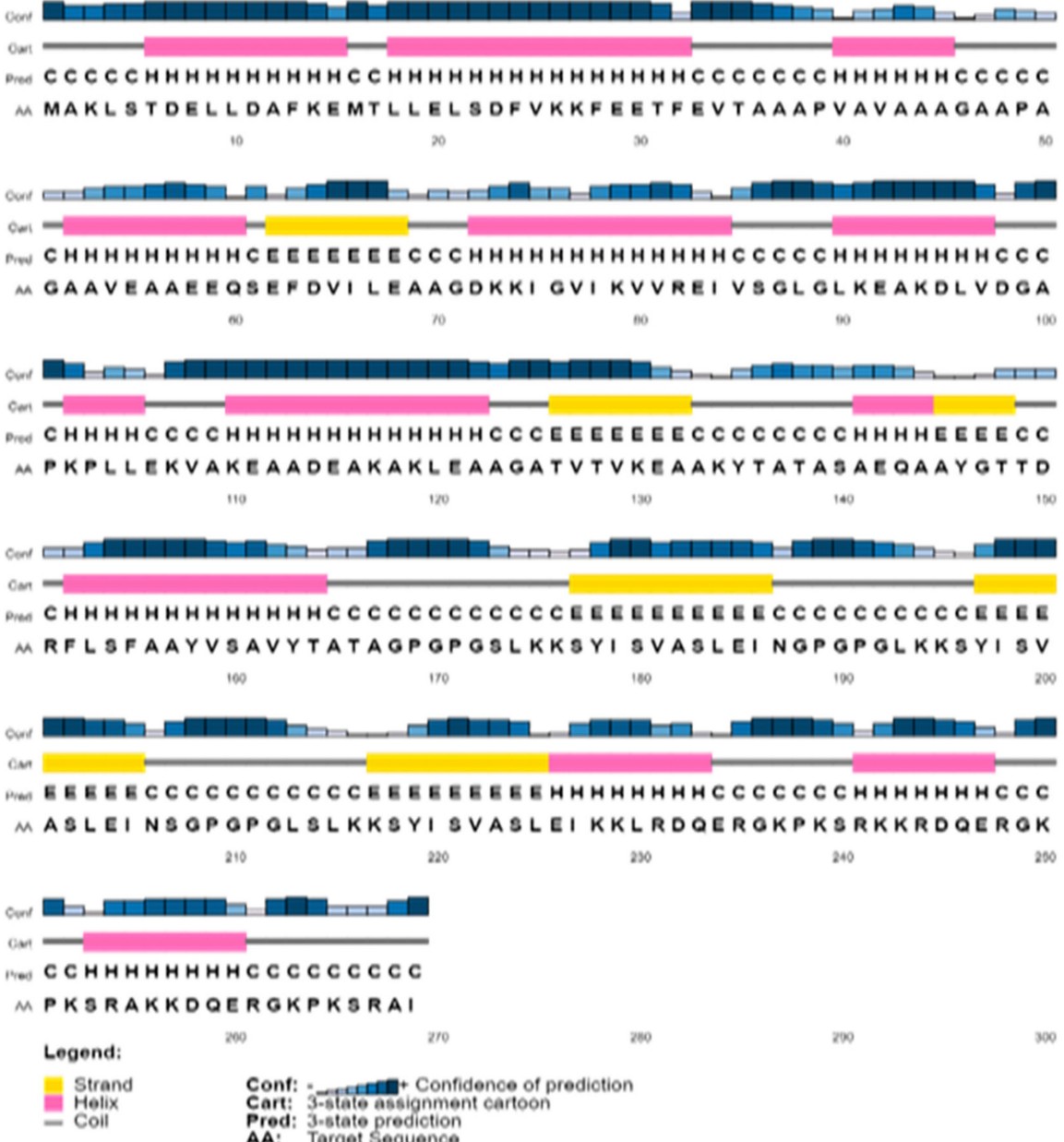

**Figure 4.** Secondary structure prediction of designed multi-epitope vaccine using PSIPRED server.

### 3.8. D Structure, Refinement, and Validation

The top five homology models were created using RaptorX as the ideal layout. We picked the model with the lowest C-score (−4.97) out of the five based on the server's recommendation. With a GDT-HA score of 0.8946, RMSD value of 0.536, MolProbity of 2697, clash score of 27.9, and poor rotamers score of 0.8, 84.5% of the vaccine's (model 1) sequences were in the favorable area in the Ramachandran plot after refining. The refined

vaccine 3D structure model was uploaded to the ProSA-web server to predict the Z-score. Before refining, the vaccine's Ramachandran plot revealed 80.7% residues in the favorable zone, 17.6% in acceptable areas, and 0.4% in prohibited regions. The Ramachandran plot of the updated vaccination model indicated 84.5% residues in the favorable zone, 14.6% residues in permitted areas, and 0.4% residues in prohibited areas (Figure 5B). The crude model had a Z-score of −5.76, whereas the refined model had a Z-score of −5.79 (Figure 5D). In addition, Figure 6 shows a structural model of the vaccine.

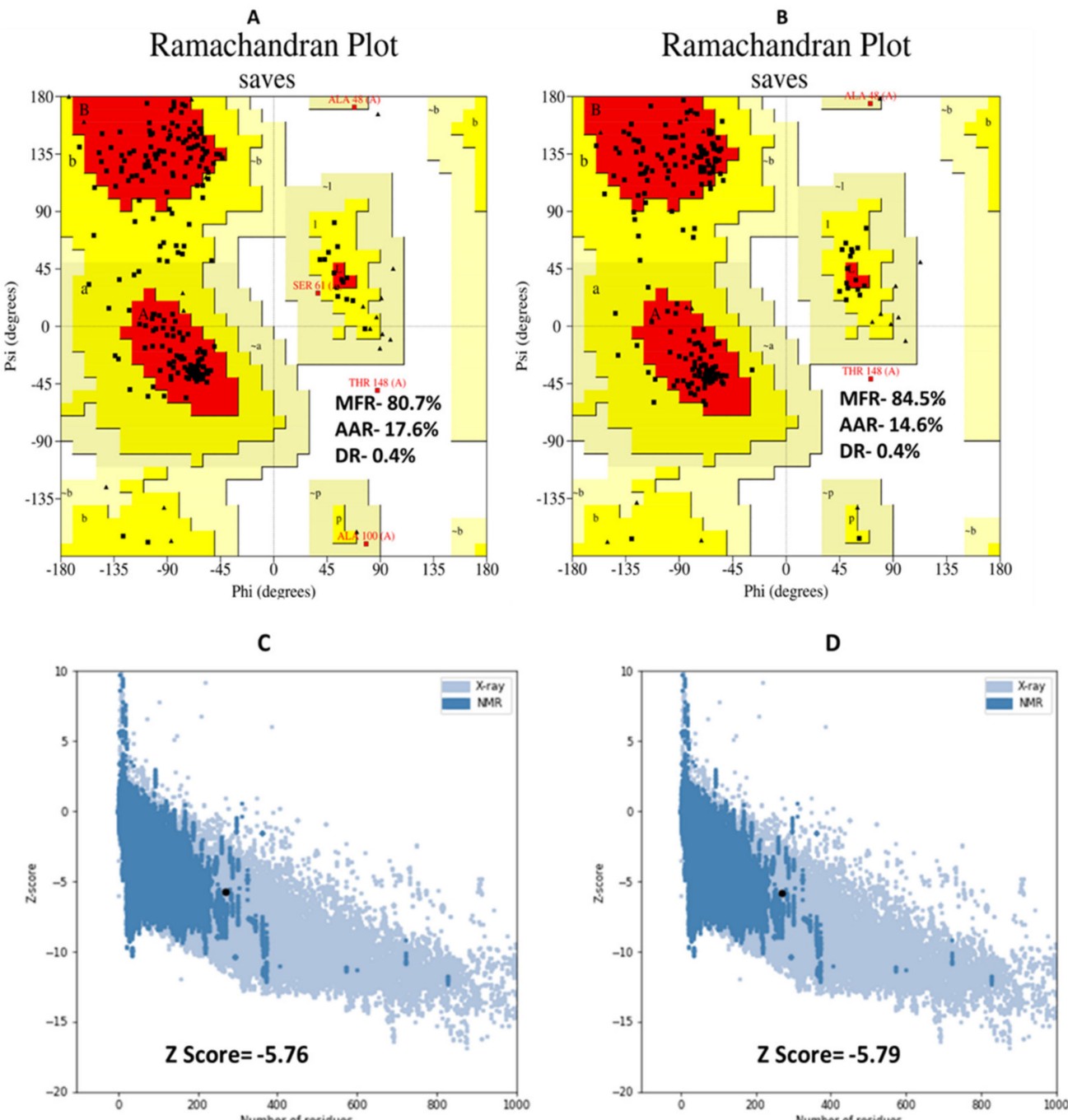

**Figure 5.** (**A,B**) PROCHECK server analysis of Ramachandran plot. The MFR, AAR, GAR, and DR sections of the vaccine were represented as the most favored, additional allowed, generously allowed, and disallowed. (**C,D**) The 3D structure validation with a Z-score by Pro-SA server.

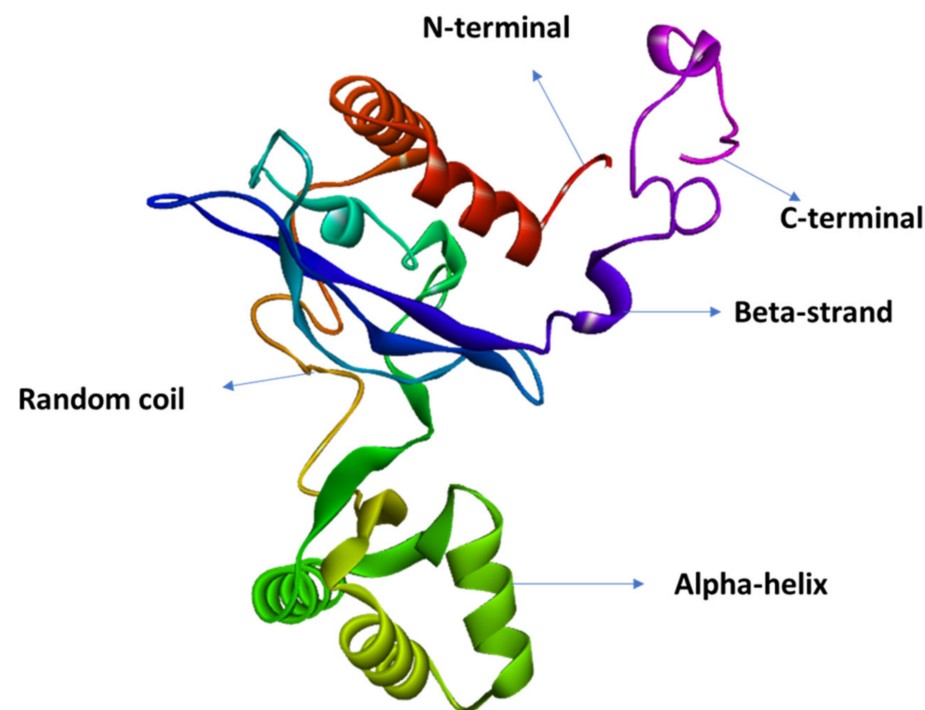

**Figure 6.** The 3D structure of the predicted vaccine construct.

*3.9. Vaccine Disulfide Engineering*

Disulfide engineering was used to stabilize the vaccine design. The DbD2 server discovered 47 pairings of amino acids with the potential to form disulfide bonds in the case of our vaccine. Nevertheless, after taking into account additional factors such as energy and the chi3 value, six pairings were selected for cysteine modification. Thus, CYS29-CYS40, LEU280-CYS306, CYS281-CYS306, CYS390-CYS391, CYS583-CYS607, and CYS585-CYS627 were the mutated residue pairs with the most mutations.

*3.10. Molecular Docking Studies*

Docking analysis was conducted between the vaccine (ligand) and TLR4 to determine their binding affinity and associations. As a result of this strategy, the ClusPro v2.0 server displayed 10 docked complexes of varied sizes; we chose the compound with the minimum energy score and interaction position with different molecules. The inclination criterion was satisfied with model 1. The best vaccine–TLR4 complex was thus selected with an energy score of −955.6. Afterward, the complex was examined for binding interactions and active site residues. The interaction surface contained 38 hydrogen bonds. Among the hydrogen bonds, there were 27 classical hydrogen bonds. The vaccine's CHB included interacting residues were Asp262, Thr213, His234, His178, Glu219, Tyr153, Thr263, Thr158, Leu232, Lys181, Lys78, Ile75, Gly211, Pro179, Val214, Ala215, Val233, Lys256, Arg278, Pro208, Lys241, Gly76, Pro210, Lys255, Lys256, Phel154, and Asp72. Figure 7 also shows the residues associated with the TLR4 active site. Six hydrogen bonds involved electrostatic salt bridges, three hydrogen bonds involved disulphide bonds, and one did not involve banded hydrogen bonds.

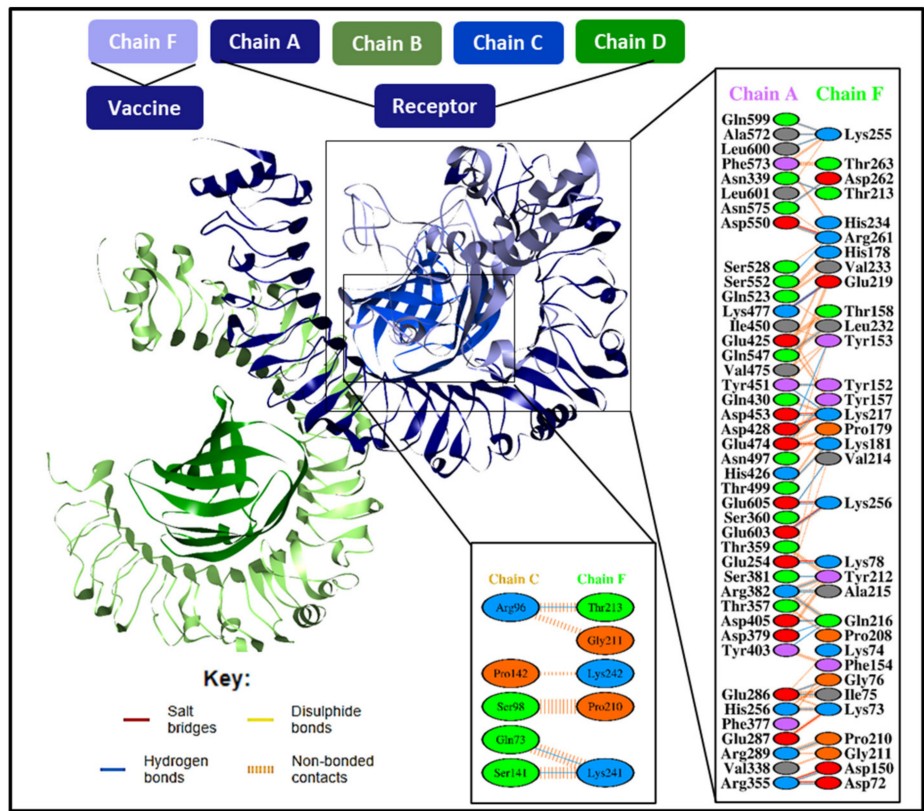

**Figure 7.** Molecular docking between the vaccine and the TLR4 receptor.

### 3.11. Molecular Dynamics Simulation

RMSD was calculated for both vaccine complexes and vaccines. Based on the RMSD value of 4.92Å, the vaccine complex showed structural stability during interaction. Figure 8 shows that the vaccine complex initially increased in RMSD descriptor, but then remained stable until 25 ns after that. A reduction in variance from 20 to 25 ns was observed, which might be important for structural integrity and/or firm binding. The root mean square fluctuation (RMSF) score was also employed to determine protein stiffness among amino acid residues. According to the RMSF profile of the vaccination complex, maximum amino acid residues from complexes with an RMSF profile below 4.0 Å and larger changes were detected for fewer residues. Figure 9 depicts the vaccine complex's stability and rigidity.

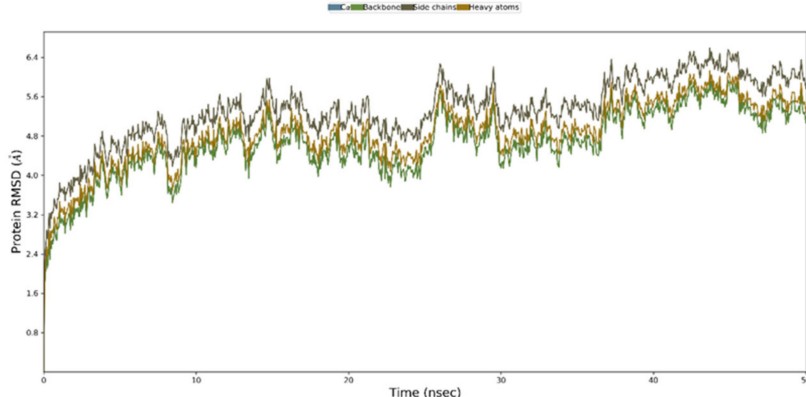

**Figure 8.** Simulation of the multi-epitope vaccination complex at the molecular level. The RMSD plot of the complexes' backbone atoms.

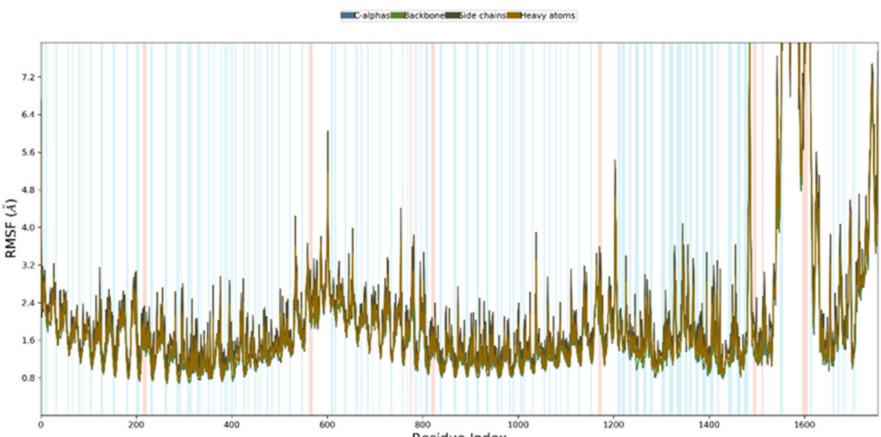

**Figure 9.** At the molecular level, simulation of the multi-epitope vaccination complex. The RMSF plot of the multi-epitope docked vaccine candidate. Sections of the α-helix and β-strand are marked in red and blue, respectively. These areas are defined by helices or strands that last for 70% of the simulation time.

### 3.12. Immune Response Simulation

The simulated immune response approximated true immunological responses triggered by some illnesses, as seen in Figure 10. For example, secondary and tertiary immune responses were larger than initial immune responses (Figure 10A). Secondary and tertiary reactions were also observed which were linked to greater antibody levels, resulting in considerably enhanced antigen clearance following successive exposures (Figure 10A). This prediction is based on higher vertebrate immune responses. B-cells, cytotoxic T-cells, and helper T-cells also showed a prolonged survival duration, indicating IgM memory formation and immune cell class switching (Figure 10B–D).

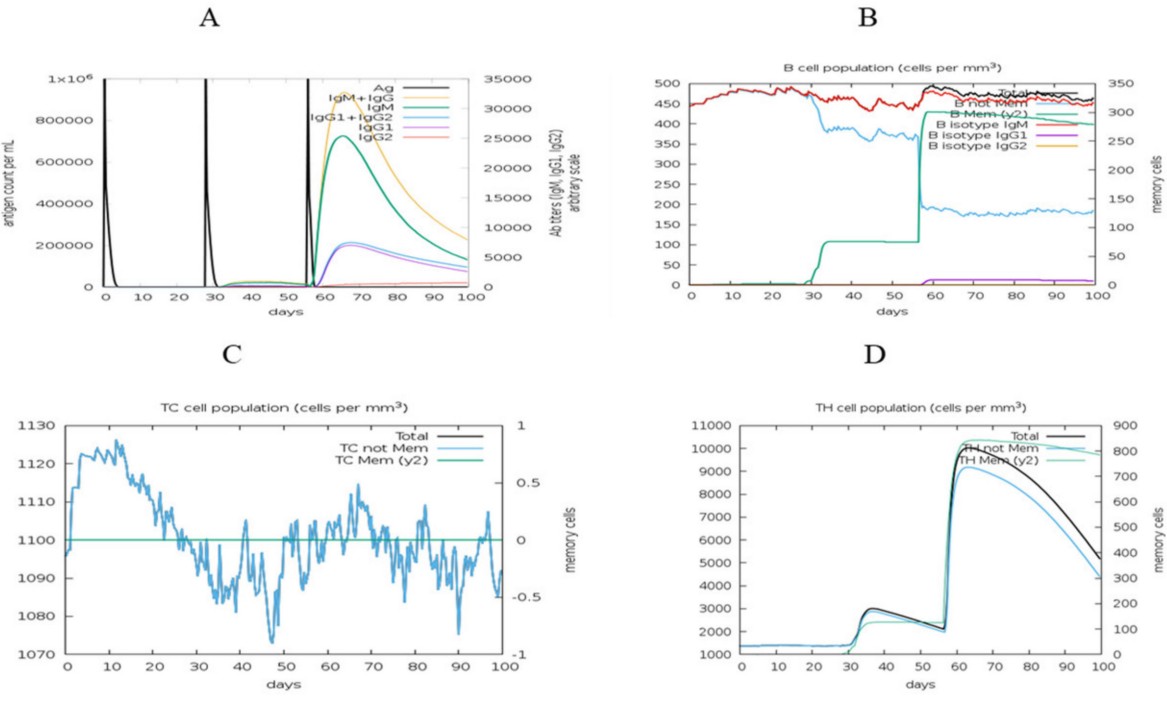

**Figure 10.** *Cont.*

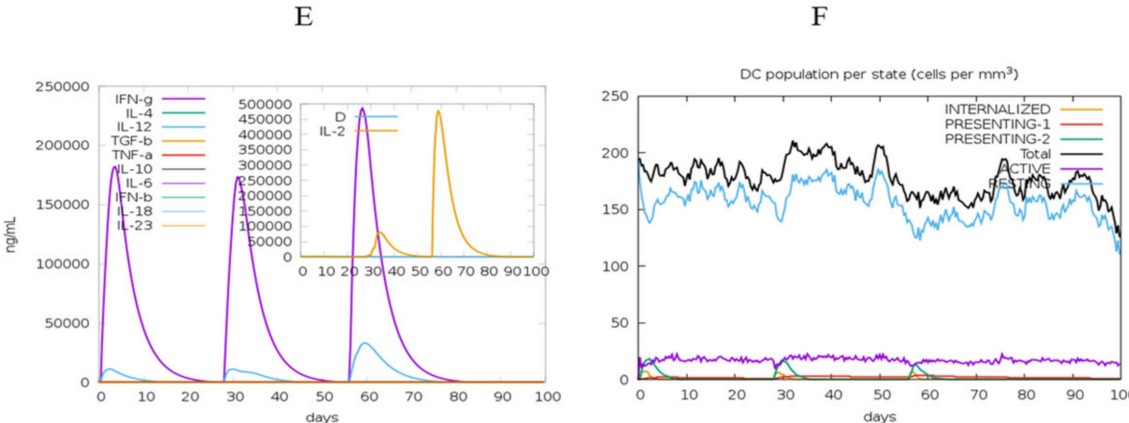

**Figure 10.** The proposed vaccination elicits an immune response. The graph shows (**A**) immunological reactions (primary, secondary, and tertiary), (**B**) B-cell population, (**C**) cytotoxic T-cell population, (**D**) helper T-cell population, (**E**) induction of cytokines and interleukins, and (**F**) dendritic cell population per state.

## 4. Discussion

Tilapia lake virus (TiLV) recently emerged, posing a severe threat to the global aquaculture sector [56]. As a result, we decided to apply an immunoinformatics strategy to create this multi-epitope vaccine. The vaccine, based on putative polymerase basic 1 (PB1), shows exceptional relevance as predicted using immunoinformatics, demonstrating the validity of our efforts. The PB1 protein is the major virulence gene that can cause TiLV in tilapia [1]. A vaccine provides safe and efficient protection against infectious diseases [57]. It should be feasible to build immunity to contagious diseases through a vaccine [58]. As a consequence of our research, we developed an epitope-based vaccination to induce a high immune response to TiLV. TiLV transmission and distribution are hard to regulate in the absence of an effective vaccine. Furthermore, an effective vaccine is yet to be ordered to control the current condition. As a result, finding a solution to the current economically damaging aquaculture problem will require a revolutionary vaccine development technique. Our objective was to design an epitope vaccine that targeted the PB1 of TiLV, which is critical for immunological invasion and fish-to-fish transmission. The antigenic area of the PB1 surface was assessed to allow cellular and humoral immune systems to identify this protein. Identifying all probable CTL, HTL, and LBL epitopes was the initial step. As the linkers below matched the top three epitopes, vaccines were created containing three antigenic epitopes—CTL, HTL, and LBL. They were employed in the development of our peptide vaccine as an essential component that increases stability, folding, and transcriptional regulation [59]. EAAAK linkers were utilized to link the adjuvant to the epitope to elicit high amounts of both cellular and immunogenic humoral responses, as well as to extend the vaccine's life and stability [60]. The vaccine's manufacture had a total of 269 amino acid residues. Solubility, a physicochemical property, is an important feature of a recombinant vaccine [61]. The vaccine design was tested using a solubility assessment tool to see if it was solvable inside the host *E. coli*; the findings revealed that it was. The vaccine's nature was acidic, as shown by the predicted PI value. According to server tools, the protein's stability index shows that it will be stable after synthesis. Based on the predicted physicochemical characteristics and scores on all metrics, this vaccine has a strong chance of becoming a viable option against TiLV. After the 3D structure prediction, the identified models were reviewed, and the best model (based on the lowest energy score) was selected (based on c-score). According to the Ramachandran plot validation test, we found a fair number of Z-scores (−5.79) and, in many cases, superior characteristics of most favored, acceptable, and prohibited areas. The vaccine will have an infection-inhibitory action and interact tightly with the TLR4 receptor, according to the minimum energy score of 955.6 for docking studies, between the peptide vaccination and the viral glycoprotein binding the receptor of TLR4.

The molecular dynamics simulation has the potential to help researchers better understand how proteins work and how their structure is created. Protein dynamic simulations as a function of time can be used to imitate anatomical movement. The vaccine candidate was dynamically simulated for 50 ns, and the results were assessed using the RMSD and RMSF scores. The RMSD value is used to evaluate two atomic conformational changes in a complex. Considerable flexibility and departure of vaccination candidates were detected using RMSD values. Atoms of the vaccine candidate were measured for their displacement from the receptor structure using the RMSF values of the complex structure. The average RMSD and RMSF values calculated were 4.92 and 4.0, respectively. The variance was less in the vaccine area, but it leveled out within 5 ns, showing that the model vaccine and receptor are stable. Finally, we looked at the ideal target clearance and cell density parameters for the best immune response to the virus using an immunological simulation. The immune system produced memory B-cells (with a half-life of many months) and T-cells as a result of the increased vaccination doses. In this way, the vaccine successfully replicated a humoral immune response to increased immunoglobulin synthesis. MD simulation was used to examine the instability of the vaccine candidate with the receptors, and nucleotide modification was conducted for the longevity of the built vaccine inside the host to increase multi-epitope vaccine generation.

## 5. Conclusions

This study employed a variety of computational algorithms to identify potential B- and T-cell epitopes in TiLV pathogenic PB1 protein, which were then stitched together into a multi-epitope vaccine. The immuno-dominant properties desired in a vaccine were just recently discovered. It might potentially bind to immunological TLR4 receptors and trigger a powerful immune response against TiLV disease. In light of our discoveries, we recommend that making an immunization against the etiological go-between of the TiLV scourge in fish ought to start with the immunization applicant. In addition, the possible epitopes discovered in this study can be employed in future studies. However, further research is needed to prove that our vaccine is an effective TiLV preventive.

**Author Contributions:** Conceptualization, S.I.I. and S.M.; methodology, S.I.I.; software, M.A.A.; validation, Y.A., S.S. and M.J.M.; formal analysis, S.M.; investigation, S.I.I.; resources, M.A.A.; data curation, S.I.I.; writing—original draft preparation, S.I.I.; writing—review and editing, M.J.M.; visualization, S.S.; supervision, S.M. All authors have read and agreed to the published version of the manuscript.

**Funding:** This research did not receive any specific grant from funding agencies in the public, commercial, or not-for-profit sectors.

**Institutional Review Board Statement:** Not applicable.

**Informed Consent Statement:** Not applicable.

**Data Availability Statement:** Not applicable.

**Acknowledgments:** The authors thank Foysal Ahmed Sagore and Kazi Abdus Samad for helpful comments.

**Conflicts of Interest:** The authors declare no conflict of interest.

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
