# Peer review of "Immunoinformatic Approaches to Identify Immune Epitopes and Design an Epitope-Based Subunit Vaccine against Emerging Tilapia Lake Virus (TiLV)"

_2673-9496, doi:10.3390/aquacj2020010_

Round 1

Reviewer 1 Report

The authors describe a bioinformatics-based approach for producing a multiepitope vaccine capable of inducing an immune response which can protect against TiLV. Though most of the servers and tools are specifically designed for higher vertebrates, the study introduces a thorough bioinformatic approach for evaluating a peptide or epitope. However, testing the resulting vaccine candidate in a fish study will be very relevant and will prove the applicability of these approaches in the future.

Title – The approach described here best suits a peptide vaccine / subunit vaccine / epitope vaccine. Nothing related to mRNA vaccine is described in the study. The title may be changed accordingly.

Line 20 specispecies’ntial Typo?

Line 40 10to0% typo?

Line 45 change the reference format

Line 56 against bacteria? Or pathogen?

line 57 A vaccination – against TiLV or aquatic viruses

Line 68 cord or Cod?

Line 90 Directly infecting - is it right?

Line 92, 114 chosen protein - mention PB1

Line 118 mention accession number PDB DOI

Line 348 IgG is not present in fish. Specify that this prediction is based on human/higher vertebrate immune responses.  

Reviewer 2 Report

To be frank, this manuscript is very interesting. It is really a hot topic right now in human beings infectious disease control, such as COVID-19. The author focused on TiLV, an emerging aquatic virus, and presented a completely new approach in designing an in-silico vaccine for the effective control of this fish disease. The choice of the research is innovative, the methods used are reasonable and the results are reliable. Moreover, the manuscript is well-formatted and well-written in professional language. A minor comment for the author is: why did the author choose PB1 gene for vaccine design other than an antigenic protein of TiLV(It is suggested that the author discuss this issue in the Part of Discussion)? In addition, please check line 40 (10to0%) and Line 42 (beenbeingssitified) carefully.

Author Response

Please find the response below in the word file.
